# Synthesis and Antifungal Activity of New butenolide Containing Methoxyacrylate Scaffold

**DOI:** 10.3390/molecules27196541

**Published:** 2022-10-03

**Authors:** Qian Zhang, Yihao Li, Bin Zhao, Leichuan Xu, Haoyun Ma, Mingan Wang

**Affiliations:** Innovation Center of Pesticide Research, Department of Applied Chemistry, China Agricultural University, Beijing 100193, China

**Keywords:** butenolide, methoxyacrylate, synthesis, antifungal activity

## Abstract

In order to improve the antifungal activity of new butenolides containing oxime ether moiety, a series of new butenolide compounds containing methoxyacrylate scaffold were designed and synthesized, based on the previous reports. Their structures were characterized by ^1^H NMR, ^13^C NMR, HR-MS spectra, and X-ray diffraction analysis. The in vitro antifungal activities were evaluated by the mycelium growth rate method. The results showed that the inhibitory activities of these new compounds against *Sclerotinia sclerotiorum* were significantly improved, in comparison with that of the lead compound **3–8**; the EC_50_ values of **V**-**6** and **VI**-**7** against *S. sclerotiorum* were 1.51 and 1.81 mg/L, nearly seven times that of **3–8** (EC_50_ 10.62 mg/L). Scanning electron microscopy (SEM) and transmission electron microscopy (TEM) observation indicated that compound **VI-3** had a significant impact on the structure and function of the hyphal cell of *S. sclerotiorum* mycelium and the positive control trifloxystrobin. Molecular simulation docking results indicated that the introduction of methoxyacrylate scaffold is beneficial to improving the antifungal activity of these compounds against *S. sclerotiorum*, which can be used as the lead for further structure optimization.

## 1. Introduction

In agricultural development, there is a long history of using pesticides to protect crops from external stress, including phytopathogen invasion, vicious weed growth and pest destruction. The large quantity use of pesticides has greatly increased the crop yields, which made a major contribution to solve the food crisis issue brought about by the world population growth [1]. Unfortunately, it also has raised some urgent concerns. For example, the resistance problems of phytopathogens, pests and weeds to the traditional pesticides are becoming more and more serious alongside the uncontrolled use of pesticides [2,3]. The commercial strobilurin fungicides have also faced this similar issue in recent years [4].

Strobilurins, the most important class of agricultural fungicides, mimicked the structure of naturally occurring compounds isolated from several basidiomycete species that inhabit decaying material in woodland soil [5,6,7]; therefore, they have been widely used as agricultural fungicides in many countries [8,9]. These compounds have a similar scaffold, the (*E*)*-*β-methoxyacrylate pharmacophore, for example, trifloxystrobin (TRI), as a member of the Qo inhibitor group, acts as the inhibitor of mitochondrial respiration system by blocking electron transfer at the ubiquinol oxidation center (Qo site) of the cytochrome bc1 complex (complex III) [10]. It has been demonstrated that the (*E*)-β-methoxyacrylate core is the basic pharmacophore of strobilurin fungicides, which, as well as its attachment to a structurally diverse side chain, is an effective way to find novel candidates with high fungicidal activities [11,12]. Recently, several studies have demonstrated that the repeated field application of strobilurin fungicides has resulted in the resistance development of several important phytopathogens [13,14]. As a consequence, new types of fungicides must be highly required and developed to overcome this problem.

Natural products are one of the important resources in the discovery of novel pesticides [15,16,17,18,19]. Butenolide scaffold is widely found in the natural products [20,21], such as stypolactone [22], lambertellol A [23], yaoshanenolides A, yaoshanenolides B [24,25] and others. These natural products have insecticidal [26,27,28], bactericidal [29,30,31], antifungal [32,33], antitumor and other excellent biological activities. The commercially available butenolide pesticides include spirodiclofen [34], spiromesifen [35], spirotetramat [36], spiropidion [37] and flupyradifurone [38]. Butenolide scaffold is a valuable antifungal pharmacophore [39,40].

The synthesis and biological activity of new butenolide compounds with the structure diversity were explored based on the diversity-oriented synthesis strategy in our research group, and found some compounds exhibit excellent insecticidal and fungicidal activities [41,42,43,44,45]. For example, some of compounds **1** and **2** had 100% mortality against *Myzus persicae*, *Mythimna seprata* and *Plutella xylostella* at the concentration of 600 mg/L [42,43], and some of compounds **3** had a 100% control effects on cucumber downy mildew and corn rust at a concentration of 400 mg/L [45] (Figure 1). When R was chlorine atom at the *ortho*-position of benzene ring in benzyl group, compound **3**–**8** exhibited the best in vitro inhibition against *Sclerotinia sclerotiorum* with EC_50_ value of 10.62 mg/L [46]. Based on the previous work, it was found that if the *ortho*-position steric hindrance of the benzene ring was increased, these compounds would exhibit much better fungicidal activities. So, a new type of butenolide derivatives were designed using the (*E*)-*β*-methoxy- acrylate pharmacophore to replace the chlorine atom at the *ortho*-position of the benzene ring and were synthesized (Figure 2), and their in vitro antifungal activities were evaluated in this article.

## 2. Results and Discussion

### 2.1. Chemistry

In order to obtain high yields of the key intermediate oximes **IV-1**~**IV-14**, 1.2 eq of NH_2_OH·HCl was added to complete the reaction (Figure 3). The single configuration isomer of the oximes could be afforded by recrystallization with ethyl acetate or CH_2_Cl_2_ at 0 °C, and the configuration of the oxime was assured as *E*-configuration by the X-ray diffraction analysis of compound **IV**-**6** (Figure 1). It was found that the best reaction condition for preparing **V-1**~**V-14** was anhydrous acetonitrile as solvent, NaH as the base, and PEG-400 as the phase transfer catalyst after repeated trials, as in the previous reports [45,46], but the yields of **V-1**~**V-14** were only 18–36% due to the bigger steric hindrance of the methoxyacrylate pharmacophore. Compounds **VI-1**~**VI-14** were readily afforded in 75–88% yields by the aminolysis of **V-1**~**V-14** using the methylamine aqueous solution. In the ^1^H NMR and ^13^C NMR spectra of **V-1**~**V-14**, it could be clearly observed that there was characteristic OCH_3_ signals at δ~4.00 and 63.8–63.9, while the NH protons presented at δ 6.70~6.90, NCH_3_ at δ 2.80~2.90 and 33.5–33.8 in the ^1^H NMR and ^13^C NMR spectra of **VI-1**~**VI-14**. The [M + H]^+^ quasi-molecular ions were observed in the high resolution mass spectra of **V-1**~**V-14** and **VI-1**~**VI-14**, for example, the [M + H]^+^ ion at m/z 480.2131 was afforded for compound **VI-8**.

### 2.2. The X-ray Structure Analysis of **IV-6**

A colorless crystal of (*E*)-3-(1-(hydroxyimino)ethyl)-5,5-dimethyl-4-phenylfuran- 2(5H)-one **IV**-**6,** suitable for X-ray diffraction analysis, was obtained from a slowly evaporating ethyl acetate solution. A 0.55 × 0.34 × 0.02 mm^3^ crystal was selected and mounted on a Bruker APEX-II CCD diffractometer, equipped with a graphite- monochromatic Mo *Kα* radiation (*λ* = 0.71073 Å). The parameters for **IV**-**6** are: formula C_14_H_15_NO_3_, molecular weight 245.27, crystal system tetragonal, a = 0.88563(5) nm, b = 0.88563(5) nm, c = 3.21060(18) nm, α = 90°, β = 90°, γ = 90°, V = 2.5182(2) nm^3^, ρ = 1.294 mg/mm^3^, space group P4_1_2_1_2, Z = 8, μ (Mo Kα) = 0.091 mm^−1^, F(000) = 1040, S = 1.085. A total of 8339 reflections were collected at T = 125.50 K; 2427 were independent reflections (*R*_int_ = 0.0522). Final R indexes were *R*_1_ = 0.0655, *wR*_2_ = 0.1114 (all data). The crystal structure was solved by direct methods with SHELXS-97 and refined by full-matrix least-squares refinements based on *F*^2^ with SHELXL-97. The crystal structure is depicted in Figure 1. The parameters and structure information for compound **IV**-**6** have been deposited at the Cambridge Crystallographic Data Centre. CCDC ID 2162653 contains the supplementary crystallographic data for this paper.

### 2.3. The Antifungal Activities

The in vitro antifungal activities against *Thanatephorus cucumeris*, *Fusarium graminearum*, *Sclerotinia sclerotiorum*, *Phytophthora capsica* and *Botrytis cinerea* were evaluated by the mycelial growth rate method [44,45,46], and the results were shown in Table 1. The results indicated that all the target compounds exhibit certain in vitro antifungal activities against the five tested phytopathogens at 50 mg/L. Although the inhibitory activities of the compounds against *T. cucumeris*, *P. capsica*, *B. cinerea* and *F. graminearum* were relatively weak, the inhibitory rates of the compounds against *S. sclerotiorum* were greater than 70%. For example, compounds **V-6** (79.4%), **V-7** (86.6%), **V-10** (89.1%), **VI-3** (74.2%), **VI-5** (75.3%), **VI-7** (79.1%) and **VI-****10** (84.8%) had better in vitro antifungal activity than TRI (68.6%). In addition, compounds, **V-10** (60.5%) and **VI-9** (69.2%), also showed good antifungal activity against *T. cucumeris*, which was close to TRI (69.8%), compound **VI-10** (60.2%) showed better antifungal activity against *P. capsica*, compounds **V-5** (49.8%) and **V-6** (46.2%) are close to that of TRI (49.7%) against *B. cinerea*, compounds **V-10** (45.4%), **VI-1** (54.8%), **VI-2** (52.5%), **VI-3** (46.3%) and **VI-4** (52.5%) were more active than TRI (43.9%) against *F. graminearum*. The preliminary structure–activity relationship indicated that the 4-position of butenolide was modified with 4-pyridinyl group; the in vitro antifungal activity was not significantly improved, in comparison with that of the compound with 4-phenyl group; and, in particular, the antifungal activity was significantly decreased for *S. sclerotiorum*. In addition, it was found that when the 5-spirocycle was replaced by a geminal dimethyl group, the in vitro antifungal activities of the compounds **V-7**, **V-8**, **V-9** and **V-10** against the five fungi were higher than those of **V-1**, **V-2**, **V-3** and **V-4** due to the steric hindrance of the spirocycle. Moreover, the EC_50_ values of compounds **V-6**, **V-7**, **V-10**, **VI-3**, **VI-5**, **VI-7** and **VI-10** against *S. sclerotiorum* were further determined and shown in Table 2, which indicates that the in vitro antifungal activities of the aminolysis products against *S. sclerotiorum* have been significantly improved. For example, the antifungal activities of compounds **VI-3** and **VI-5** were much higher than those of **V-3** and **V-5**. However, there was some exceptions, the antifungal activities of **V-6** and **V-10** were better than those of **VI-6** and **VI-10**. These results also showed that the EC_50_ values of **V-6**, **V-7**, **V-10**, **VI-3**, **VI-5**, **VI-7** and **VI-10** (1.51–13.24 mg/L) were significantly smaller than that of TRI (38.09 mg/L), the splicing of butenolide and methoxyacrylate is beneficial to improve the in vitro antifungal activity of these compounds. In addition, it was confirmed that the antifungal activities of compounds against *S. sclerotiorum* was significantly improved from 10.62 mg/L of **3–8** to 1.51 and 1.81 mg/L of **V**-**6** and **VI**-**7** by increasing the steric hindrance of the *ortho*-position in the benzene ring.

In addition, the effects of compound **VI-3** on the hyphae and cell morphology of *S. sclerotiorum*, and the difference between **VI-3**, TRI and the blank control were observed. By TEM observation, the hyphal cells were normal, the cell membrane was intact, the cytoplasmic organelles were evenly dispersed, the large vacuoles existed for the hyphae of the blank control. Whereas the ultrastructure of the hyphal changed dramatically, the inner mitochondria of the cells swelled and became disordered; the folds of mitochondrial cristae disappeared or the vacuoles were no longer evenly distributed, some of them even ruptured and became smaller after compound **VI-3** and trifloxystrobin treatment in *S. sclerotiorum* cell (Figure 2**,** Figure S83 in Appendix A). The similar symptom, such as the folds of mitochondrial cristae disappearance, was observed for the another strobilurin fungicide azoxystrobin, which also act as the inhibitor of mitochondrial respiration system by blocking electron transfer at the ubiquinol oxidation center (Qo site) of the cytochrome bc1 complex [47]. The irregular bifurcation and bulge at the hyphal of *S. sclerotiorum* were found for the hyphae treated with TRI and **VI-3** (Figure 3, Appendix A in Appendix A) by observing SEM. Based on the above results, it can be deduced that the mitochondria and vacuoles in the hyphal cells may be damaged, and TRI and **VI-3** should have a similar mechanism of action as azoxystrobin.

### 2.4. The Docking Study

In order to explore the possible mode of action for this type of compounds and their similar mechanism with TRI, compounds **V**-**3** with low activity and **V**-**6** and **VI**-**7** with high activity were selected for molecular docking with the target protein (PDB: 1SQB) [48] and comparison with TRI. The results showed that the H-bond interaction with a distance of 2.2 Å both **V**-**6** and **VI**-**7** with the key Glu 271 amino acid residue in the active cavity of the protein was observed while there was no H-bond interaction in either TRI or **V**-**3** with the key amino acid residues in the active cavity because of the bigger steric hindrance of spirobutenolide moiety and *tert*-butylphenyl group in **V**-**3**. These differences might rationalize the antifungal activity differences of **V**-**3**, **V**-**6** and **VI**-**7**. From the molecular docking point of view, the O atom in the methoxyacrylate scaffold is the key atom to form the H-bond with the target molecule for improving the antifungal activity, that make **V**-**6** and **VI**-**7** more active than TRI and **V**-**3** (Figure 4, Figure S84 in Appendix A).

## 3. Materials and Methods

### 3.1. General Information

All reactions were performed with magnetic stirring. Unless otherwise stated, all reagents were purchased from commercial suppliers (Energy chemical, Shanghai, China) and used without further purification. Organic solutions were concentrated under reduced pressure using a rotary evaporator or oil pump. WRS-3 Micro Melting Point Apparatus (uncorrected), Bruker DPX 500 MHz NMR instruments, TMS as the internal standard, CDCl_3_ as the solvent, chemical shifts are represented as δ, Agilent 1100 LC-MSD-Trap mass spectrometer (ESI-MS), Cold Field-Emission SU-8010 Scanning Electron Microscope, Hitachi-7500 Transmission electron microscope, Thermo Fisher MSQ-Plus high-resolution mass spectrometer, Thermo Fisher ESCALAB 250 four-circle X-ray diffractometer. The organic solutions were concentrated using a rotary evaporator. The silica gel (200–300 mesh, Qingdao Haiyang Chemical Co., Ltd., Qingdao, China) was used for column chromatography. The reagents were analytical grade, and the anhydrous solvent was analytical grade and dried by conventional methods.

### 3.2. The Synthesis of Intermediates **III-1****~III-14** and **IV-1****~IV-14**

The compounds **I-1**~**I-8**, **II-1**~**II-8**, **III-1**~**III-8** and **IV-1**~**IV-8** were synthesized in the previous report [46]. The compounds **I-9**~**I-14** and **II-9**~**II-14** were synthesized following the procedures in the literature [49,50], the key intermediates **III-9**~**III-14** and **IV-9**~**IV-14** were synthesized following the protocols in the previous report [41,42,43,44,45,46].

*3-Acetyl-4-(4-fluorophenyl)-5,5-dimethylfuran-2(5H)-one* (**III**-**9**): White solid, Rf 0.57 (*V*_ethyl acetate_: *V*_petroleum ether_ = 1: 5), 2.92 g, yield 82%, m.p.71–73 °C; ^1^H NMR (500 MHz, CDCl_3_) *δ*: 7.28–7.22 (m, 2H), 7.19–7.14 (m, 2H), 2.44 (s, 3H), 1.56 (s, 6H); ^13^C NMR (126 MHz, CDCl_3_) *δ*: 194.5, 175.3, 168.5, 163.7 (d, ^1^*J_FC_* = 249.5 Hz), 129.3 (d, ^3^*J_FC_* = 8.1 Hz), 126.8 (d, ^4^*J_FC_* = 3.4 Hz), 126.7, 116.3(d, ^2^*J_FC_* = 22.5 Hz), 86.4, 30.6, 24.9; HR-MS (ESI) *m*/*z*: C_14_H_14_FO_3_[M + H]^+^, Calcd. 249.0921, Found 249.0925.

*3-Acetyl-4-(4-methoxyphenyl)-5,5-dimethylfuran-2(5H)-one (***III**-**10**): White solid, Rf 0.54 (*V*_ethyl acetate_: *V*_petroleum ether_ = 1: 10), 2.26 g, yield 81%, m.p.71–72 °C; ^1^H NMR (500 MHz, CDCl_3_) *δ*: 7.25–7.21 (m, 2H), 6.99–6.94 (m, 2H), 3.85 (s, 3H), 2.41 (s, 3H), 1.57 (s, 6H); ^13^C NMR (126 MHz, CDCl_3_) *δ*: 195.3, 175.5, 168.8, 161.3, 129.2, 125.8, 122.8, 114.4, 86.4, 55.5, 30.6, 25.3; HR-MS (ESI) *m*/*z*: C_15_H_17_O_4_[M + H]^+^, Calcd. 261.1121, Found 261.1126.

*3-Acetyl-4-(4-(tert-butyl)phenyl)-5,5-dimethylfuran-2(5H)-one* (**III**-**11**): White solid, Rf 0.62 (*V*_ethyl acetate_: *V*_petroleum ether_ = 1: 5), 1.47 g, yield 74%, m.p.69–70 °C; ^1^H NMR (500 MHz, CDCl_3_) *δ*: 7.47 (d, *J* = 8.4 Hz, 2H), 7.19 (d, *J* = 8.4 Hz, 2H), 2.41 (s, 3H), 1.58 (s, 6H), 1.35 (s, 9H); ^13^C NMR (126 MHz, CDCl_3_) *δ*: 195.1, 176.1, 168.8, 153.5, 127.9, 127.0, 126.4, 125.8, 86.5, 35.0, 31.3, 30.7, 25.1; HR-MS (ESI) *m*/*z*: C_18_H_23_O_3_[M + H]^+^, Calcd. 287.1642, Found 287.1645.

*3-Acetyl-4-(4-bromophenyl)-5,5-dimethylfuran-2(5H)-one* (**III**-**12**): White solid, Rf 0.68 (*V*_ethyl acetate_: *V*_petroleum ether_ = 1: 5), 1.43 g, yield 73%, m.p.70–71 °C; ^1^H NMR (500 MHz, CDCl_3_) *δ*: 7.62–7.56 (m, 2H), 7.13–7.07 (m, 2H), 2.44 (s, 3H), 1.53 (s, 6H); ^13^C NMR (126 MHz, CDCl_3_) *δ*: 194.2, 175.2, 168.4, 132.1, 129.8, 128.6, 126.7, 124.5, 86.2, 30.5, 24.8; HR-MS (ESI) *m*/*z*: C_14_H_14_BrO_3_[M + H]^+^, Calcd. 309.0121, Found 309.0122.

*3-Acetyl-5,5-dimethyl-4-(pyridin-2-yl)furan-2(5H)-one* (**III**-**13**): White solid, Rf 0.62 (*V*_ethyl acetate_: *V*_petroleum ether_ = 1: 5), 1.52 g, yield 97%, m.p. 225–227 °C; ^1^H NMR (500 MHz, CDCl_3_) *δ*: 8.67 (d, *J* = 4.7 Hz, 1H), 7.75 (td, *J* = 7.8, 1.8 Hz, 1H), 7.51 (d, *J* = 7.9 Hz, 1H), 7.36–7.31 (m, 1H), 2.49 (s, 3H), 1.68 (s, 6H); ^13^C NMR (126 MHz, CDCl_3_) *δ*: 196.4, 170.1, 168.6, 149.8, 149.3, 136.6, 127.8, 125.1, 124.8, 87.0, 30.7, 25.7; HR-MS (ESI) *m*/*z*: C_13_H_14_NO_3_ [M + H]^+^, Calcd. 232.0968, Found 232.0966.

*3-Acetyl-5,5-dimethyl-4-(pyridin-4-yl)furan-2(5H)-one* (**III**-**14**): White solid, Rf 0.67 (*V*_ethyl acetate_: *V*_petroleum ether_ = 1: 5), 1.57 g, yield 87%, m.p.229–231 °C; ^1^H NMR (500 MHz, CDCl_3_) *δ*: 8.72 (d, *J* = 6.0 Hz, 2H), 7.11 (d, *J* = 6.0 Hz, 2H), 2.49 (s, 3H), 1.54 (s, 6H); ^13^C NMR (126 MHz, CDCl_3_) *δ*: 193.5, 173.7, 168.0, 150.2, 139.5, 126.9, 121.3, 86.0, 30.3, 24.6; HR-MS (ESI) *m*/*z*: C_13_H_14_NO_3_[M + H]^+^, Calcd. 232.0968, Found 232.0964.

*(E)-4-(4-fluorophenyl)-3-(1-(hydroxyimino)ethyl)-5,5-dimethylfuran-2(5H)-one* (**IV**-**9**): White solid, Rf 0.51 (*V*_ethyl acetate_: *V*_petroleum ether_ = 1: 5), 2.24 g, yield 83%, m.p.209–210 °C; ^1^H NMR (500 MHz, DMSO-*d*_6_) *δ*: 11.34 (s, 1H), 7.44–7.38 (m, 2H), 7.32 (t, *J* = 8.9 Hz, 2H), 1.84 (s, 3H), 1.51 (s, 6H); ^13^C NMR (126 MHz, DMSO) *δ*: 169.6, 167.5, 164.0 (d, ^1^*J_FC_* = 246.3 Hz), 147.4, 130.7 (d, ^3^*J_FC_* = 8.4 Hz), 128.3 (d, ^4^*J_FC_* = 3.2 Hz), 125.3, 116.3 (d, ^2^*J_FC_* = 21.3 Hz), 86.5, 40.3, 40.2, 40.0, 39.8, 39.6, 25.3, 14.0; HR-MS (ESI) *m*/*z*: C_14_H_15_FNO_3_[M + H]^+^, Calcd. 264.1030, Found 264.1034.

*(E)-3-(1-(hydroxyimino)ethyl)-4-(4-methoxyphenyl)-5,5-dimethylfuran-2(5H)-one* (**IV**-**10**): White solid, Rf 0.64 (*V*_ethyl acetate_: *V*_petroleum ether_ = 1: 5), 2.08 g, yield 72%, m.p. 188–189 °C; ^1^H NMR (500 MHz, CDCl_3_) *δ*: 9.63 (s, 1H), 7.24–7.20 (m, 2H), 6.97–6.93 (m, 2H), 3.85 (s, 3H), 1.78 (s, 3H), 1.56 (s, 6H); ^13^C NMR (126 MHz, CDCl_3_) *δ*: 169.8, 168.9, 160.9, 149.2, 129.4, 123.7, 123.6, 114.5, 86.1, 55.5, 25.6, 14.0; HR-MS (ESI) *m*/*z*: C_15_H_18_NO_4_[M + H]^+^, Calcd. 276.1230, Found 276.1235.

*(E)-4-(4-(tert-butyl)phenyl)-3-(1-(hydroxyimino)ethyl)-5,5-dimethylfuran-2(5H)-one* (**IV**-**11**): White solid, Rf 0.56 (*V*_ethyl acetate_: *V*_petroleum ether_ = 1: 5), 1.65 g, yield 71%, m.p. 174–175 °C; ^1^H NMR (500 MHz, CDCl_3_) *δ*: 8.87 (s, 1H), 7.44 (d, *J* = 8.3 Hz, 2H), 7.20 (d, *J* = 8.3 Hz, 2H), 1.81 (s, 3H), 1.57 (s, 6H), 1.34 (s, 9H); ^13^C NMR (126 MHz, CDCl_3_) *δ*: 169.6, 169.3, 153.1, 149.4, 128.5, 127.4, 125.8, 123.9, 86.2, 34.8, 31.2, 25.3, 13.8; HR-MS (ESI) *m*/*z*: C_18_H_24_NO_3_ [M + H]^+^, Calcd. 302.1751, Found 302.1754.

*(E)-4-(4-bromophenyl)-3-(1-(hydroxyimino)ethyl)-5,5-dimethylfuran-2(5H)-one* (**IV**-**12**): White solid, Rf 0.62 (*V*_ethyl acetate_: *V*_petroleum ether_ = 1: 5), 1.21 g, yield 87%, m.p. 225–227 °C; ^1^H NMR (500 MHz, CDCl_3_) *δ*: 8.46 (s, 1H), 7.59 (d, *J* = 8.5 Hz, 2H), 7.13 (d, *J* = 8.5 Hz, 2H), 1.88 (s, 3H), 1.54 (s, 6H); ^13^C NMR (126 MHz, CDCl_3_) *δ*: 169.2, 167.7, 149.2, 132.1, 130.5, 129.3, 124.2, 86.0, 25.2, 13.7; HR-MS (ESI) *m*/*z*: C_14_H_15_BrNO_3_[M + H]^+^, Calcd. 324.0230, Found 324.0236.

*(E)-3-(1-(hydroxyimino)ethyl)-5,5-dimethyl-4-(pyridin-2-yl)furan-2(5H)-one* (**IV-13**): White solid, Rf 0.57 (*V*_ethyl acetate_: *V*_petroleum ether_ = 1: 5), 1.28 g, yield 80%, m.p. 150–151 °C; ^1^H NMR (500 MHz, CDCl_3_) *δ*: 9.81 (brs, 1H), 8.73–8.64 (m, 1H), 7.74 (td, *J* = 7.8, 1.7 Hz, 1H), 7.44 (dd, *J* = 7.9, 0.9 Hz, 1H), 7.38–7.29 (m, 1H), 1.91 (s, 3H), 1.68 (s, 6H); ^13^C NMR (126 MHz, CDCl_3_) *δ*: 169.9, 165.9, 150.5, 149.9, 149.0, 136.7, 125.4, 125.2, 124.6, 124.4, 87.2, 25.9, 14.3; HR-MS (ESI) *m*/*z*: C_13_H_15_N_2_O_3_[M + H]^+^, Calcd. 247.1077, Found 247.1076.

*(E)-3-(1-(hydroxyimino)ethyl)-5,5-dimethyl-4-(pyridin-4-yl)furan-2(5H)-one* (**IV**-**14**): White solid, Rf 0.56 (*V*_ethyl acetate_: *V*_petroleum ether_ = 1: 5), 1.54 g, yield 98%, m.p. 182–184 °C; ^1^H NMR (500 MHz, CDCl_3_) *δ*: 11.26 (s, 1H), 8.46 (d, *J* = 6.0 Hz, 2H), 7.17 (d, *J* = 6.0 Hz, 2H), 2.12 (s, 3H), 1.54 (s, 6H); ^13^C NMR (126 MHz, CDCl_3_) *δ*: 169.4, 164.7, 149.1, 148.0, 141.7, 125.8, 122.9, 85.8, 25.1, 12.8; HR-MS (ESI) *m*/*z*: C_13_H_15_N_2_O_3_[M + H]^+^, Calcd. 247.1077, Found 247.1075.

### 3.3. The Synthesis of Target Compounds **V-****1****~****V****I****-****14**

The synthesis of the target compounds **V**-**1**~**V**-**14** (took compound **V**-**1** as an example): to a 200 mL round bottom flask **IV-1** (0.50 g, 1.65 mmol), NaH (0.048 g, 1.98 mmol), anhydrous acetonitrile 60 mL and 1 mL PEG-400 were added, the mixture was stirred at room temperature for 10 h. Then, benzyl bromide (0.71 g, 2.48 mmol) was added, and stirred at room temperature for 24 h. The solvent was removed under the reduced pressure, added 30 mL of water, extracted three times using 30 mL of ethyl acetate, and combined the organic phase. The organic phase was washed with the saturated brine, dried over anhydrous Na_2_SO_4_, and the solvent was removed under the reduced pressure. The residue was subjected to silica gel column chromatography and washed with ethyl acetate: petroleum ether to afford a white solid compound **V**-**1**. Compounds **V**-**2**~**V**-**1****4** were synthesized in a similar procedure.

*Methyl (E)-2-(2-(((((E)-1-(4-(4-fluorophenyl)-2-oxo-1-oxaspiro[4.5]dec-3-en-3-yl)ethylidene)**- amino)oxy)methyl)phenyl)-2-(methoxyimino)acetate* (**V**-**1**): White solid, Rf 0.56 (*V*_ethyl acetate_: *V*_petroleum ether_ = 1: 3), 0.53 g, yield 22%, m.p.120–122 °C; ^1^H NMR (500 MHz, CDCl_3_) *δ*: 7.34 (t, *J* = 7.5 Hz, 1H), 7.28 (t, *J* = 7.5 Hz, 1H), 7.18–7.05 (m, 4H), 6.98 (t, *J* = 8.6 Hz, 2H), 4.83 (s, 2H), 4.00 (s, 3H), 3.82 (s, 3H), 1.95 (s, 3H), 1.79–1.38 (m, 9H), 1.09–1.01 (m, 1H); ^13^C NMR (126 MHz, CDCl_3_) *δ*: 170.1, 168.9, 163.4, 163.0 (d, ^1^*J*_FC_ = 247.4 Hz), 149.5, 148.9, 136.1, 130.1 (d, ^3^*J*_FC_ = 8.6 Hz), 129.7, 129.4, 128.4, 128.4, 127.6, 127.5 (d, ^4^*J*_FC_ = 3.3 Hz), 125.2, 115.4 (d, ^2^*J*_FC_ = 21.6 Hz), 88.2, 74.4, 63.9, 53.0, 33.5, 24.4, 22.0, 14.3; HR-MS (ESI) *m*/*z*: C_28_H_30_FN_2_O_6_ [M + H]^+^, Calcd. 509.2088, Found 509.2082.

*Methyl (E)-2-(methoxyimino)-2-(2-(((((E)-1-(4-(4-methoxyphenyl)-2-oxo-1-oxa**-spiro[4.5]dec-3-en-3-yl)ethylidene)amino)oxy)methyl)phenyl)acetate* (**V**-**2**): White solid, Rf 0.51 (*V*_ethyl acetate_: *V*_petroleum ether_ = 1: 3), 0.47 g, yield 22%, m.p.39–41 °C; ^1^H NMR (500 MHz, CDCl_3_) *δ*: 7.38–7.29 (m, 2H), 7.25–7.23 (m, 1H), 7.16–7.10 (m, 3H), 6.86–6.82 (m, 2H), 4.91 (s, 2H), 4.00 (s, 3H), 3.83 (s, 3H), 3.81 (s, 3H), 1.89 (s, 3H), 1.85–1.64 (m, 9H), 1.20–1.04 (m, 1H); ^13^C NMR (126 MHz, CDCl_3_) *δ*: 170.3, 169.7, 163.4, 160.3, 149.5, 149.3, 136.1, 129.7, 129.6, 129.4, 128.6, 128.3, 127.6, 123.7, 113.9, 88.3, 74.4, 63.9, 55.4, 53.0, 33.8, 24.5, 22.1, 14.7; HR-MS (ESI) *m*/*z*: C_29_H_33_N_2_O_7_ [M + H]^+^, Calcd. 521.5896, Found 521.5895.

*Methyl (E)-2-(2-(((((E)-1-(4-(4-(tert-butyl)phenyl)-2-oxo-1-oxaspiro[4.5]dec-3-en-3-yl)ethyli**- dene)amino)oxy)methyl)phenyl)-2-(methoxyimino)acetate* (**V**-**3**): White solid, Rf 0.48 (*V*_ethyl acetate_: *V*_petroleum ether_ = 1: 5), 0.26 g, yield 35%, m.p.127–128 °C; ^1^H NMR (500 MHz, CDCl_3_) *δ*: 7.39–7.29 (m, 5H), 7.17–7.09 (m, 3H), 4.89 (s, 2H), 3.98 (s, 3H), 3.77 (s, 3H), 1.84 (s, 3H), 1.80–1.65 (m, 9H), 1.33 (s, 9H), 1.16–1.05 (m, 1H); ^13^C NMR (126 MHz, CDCl_3_) *δ*: 170.2, 170.0, 163.4, 152.4, 149.5, 149.2, 136.0, 129.9, 129.4, 128.8, 128.7, 128.4, 127.8, 127.7, 125.3, 88.3, 74.5, 63.9, 53.0, 34.9, 33.7, 31.4, 24.5, 22.1, 14.7; HR-MS (ESI) *m*/*z*: C_32_H_39_N_2_O_6_ [M + H]^+^, Calcd. 547.2801, Found 547.2803.

*Mthyl (E)-2-(2-(((((E)-1-(4-(4-bromophenyl)-2-oxo-1-oxaspiro[4.5]dec-3-en-3-yl)ethylidene)**- amino)oxy)methyl)phenyl)-2-(methoxyimino)acetate* (**V**-**4**): White solid, Rf 0.58 (*V*_ethyl acetate_: *V*_petroleum ether_ = 1: 3), 0.31 g, yield 20%, m.p.132–135 °C; ^1^H NMR (500 MHz, CDCl_3_) *δ*: 7.41 (d, *J* = 8.4 Hz, 2H), 7.37 (t, *J* = 7.5 Hz, 1H), 7.31 (t, *J* = 7.6 Hz 1H), 7.13 (t, *J* = 7.5 Hz, 2H), 6.99 (d, *J* = 8.4 Hz, 2H), 4.83 (s, 2H), 4.00 (s, 3H), 3.82 (s, 3H), 1.96 (s, 3H), 1.81–1.52 (m, 9H), 1.16–1.00 (m, 1H); ^13^C NMR (126 MHz, CDCl_3_) *δ*: 170.0, 168.5, 163.4, 149.5, 148.9, 136.1, 131.5, 130.6, 129.7, 129.6, 129.4, 128.4, 128.3, 127.7, 125.2, 123.5, 88.2, 74.5, 63.9, 53.1, 33.5, 24.4, 22.0, 14.2; HR-MS (ESI) *m*/*z*: C_28_H_30_BrN_2_O_6_ [M + H]^+^, Calcd. 569.1282, Found 569.1284.

*Methyl (E)-2-(2-(((((E)-1-(4-(2-chlorophenyl)-2-oxo-1-oxaspiro[4.5]dec-3-en-3-yl)ethylidene)**- amino)oxy)methyl)phenyl)-2-(methoxyimino)acetate* (**V**-**5**): White solid, Rf 0.49 (*V*_ethyl acetate_: *V*_petroleum ether_ = 1: 3), 0.24 g, yield 21%, m.p.124–126 °C; ^1^H NMR (500 MHz, CDCl_3_) *δ*: 7.40–7.36 (m, 1H), 7.33–7.24 (m, 4H), 7.14–7.06 (m, 3H), 4.66 (s, 2H), 3.99 (s, 3H), 3.82 (s, 3H), 2.04 (s, 3H), 1.95–1.46 (m, 9H), 1.15–0.99 (m, 1H); ^13^C NMR (126 MHz, CDCl_3_) *δ*: 169.9, 166.0, 163.3, 149.4, 148.9, 136.0, 131.4, 129.9, 129.8, 129.7, 129.4, 128.4, 128.3, 127.5, 126.4, 126.3, 88.96, 74.5, 63.9, 53.1, 33.7, 33.5, 24.5, 22.1, 22.0, 13.0; HR-MS (ESI) *m*/*z*: C_28_H_30_ClN_2_O_6_ [M + H]^+^, Calcd. 525.1787, Found 525.1782.

*Methyl (E)-2-(2-(((((E)-1-(5,5-dimethyl-2-oxo-4-phenyl-2,5-dihydrofuran-3-yl)ethylidene)amino)**oxy)methyl)phenyl)-2-(methoxyimino)acetate* (**V**-**6**): White solid, Rf 0.54 (*V*_ethyl acetate_: *V*_petroleum ether_ = 1: 3), 0.52 g, yield 21%, m.p.196–198 °C; ^1^H NMR (500 MHz, CDCl_3_) *δ*: 7.46–7.29 (m, 5H), 7.25–7.19 (m, 3H), 7.14 (dd, *J* = 7.0, 2.0 Hz, 1H), 4.87 (s, 2H), 3.99 (s, 3H), 3.81 (s, 3H), 1.91 (s, 3H), 1.55 (s, 6H); ^13^C NMR (126 MHz, CDCl_3_) *δ*: 169.9, 169.3, 163.4, 149.5, 149.0, 136.0, 131.5, 129.8, 129.5, 128.7, 128.5, 128.4, 128.0, 127.7, 124.5, 86.6, 74.6, 63.9, 53.0, 25.3, 14.5; HR-MS (ESI) *m*/*z*: C_25_H_27_N_2_O_6_ [M + H]^+^, Calcd. 451.1864, Found 451.1865.

*Methyl (E)-2-(2-(((((E)-1-(4-(4-fluorophenyl)-5,5-dimethyl-2-oxo-2,5-dihydrofuran-3-yl)ethyli**- dene)amino)oxy)methyl)phenyl)-2-(methoxyimino)acetate* (**V**-**7**): Yellow liquid, Rf 0.54 (*V*_ethyl acetate_: *V*_petroleum ether_ = 1: 3), 0.62 g, yield 36%; ^1^H NMR (500 MHz, CDCl_3_) *δ*: 7.36 (t, *J* = 7.4 Hz, 1H), 7.30 (t, *J* = 7.5 Hz, 1H), 7.20–7.76 (m, 3H), 7.14 (d, *J* = 7.5 Hz, 1H), 6.98 (t, *J* = 8.6 Hz, 2H), 4.86 (s, 2H), 4.00 (s, 3H), 3.82 (s, 3H), 1.96 (s, 3H), 1.53 (s, 6H); ^13^C NMR (126 MHz, CDCl_3_) *δ*: 169.8, 168.2, 163.4, 163.3 (d, ^1^*J*_FC_ = 248.4 Hz), 149.6, 149.0, 136.1, 130.1(d, ^3^*J*_FC_ = 8.7 Hz), 129.8, 129.4, 128.6, 128.4, 127.8, 127.2 (d, ^4^*J*_FC_ = 3.3 Hz), 124.7, 115.7 (d, ^2^*J*_FC_ = 21.5 Hz), 86.5, 74.6, 63.9, 53.0, 25.3, 14.4; HR-MS (ESI) *m*/*z*: C_25_H_26_FN_2_O_6_ [M + H]^+^, Calcd. 469.1769, Found 469.1765.

*Methyl (E)-2-(methoxyimino)-2-(2-(((((E)-1-(4-(4-methoxyphenyl)-5,5-dimethyl-2-oxo-2,5-di**-hydrofuran-3-yl)ethylidene)amino)oxy)methyl)phenyl)acetate* (**V**-**8**): White solid, Rf 0.42 (*V*_ethyl acetate_: *V*_petroleum ether_ = 1: 3), 0.34 g, yield 20%, m.p.134–136 °C; ^1^H NMR (500 MHz, CDCl_3_) *δ*: 7.37–7.28 (m, 3H), 7.24–7.19 (m, 2H), 7.15 (dd, *J* = 7.5, 1.8 Hz, 1H), 6.87–6.82 (m, 2H), 4.94 (s, 2H), 4.00 (s, 3H), 3.83 (s, 3H), 3.81 (s, 3H), 1.92 (s, 3H), 1.57 (s, 6H); ^13^C NMR (126 MHz, CDCl_3_) *δ*: 170.0, 168.8, 163.4, 160.7, 149.6, 149.5, 136.1, 129.9, 129.8, 129.4, 128.8, 128.4, 127.7, 123.3, 114.1, 86.5, 74.5, 63.9, 55.4, 53.0, 25.7, 14.8; HR-MS (ESI) *m*/*z*: C_26_H_29_N_2_O_7_ [M + H]^+^, Calcd. 481.1969, Found 481.1971.

*Methyl (E)-2-(2-(((((E)-1-(4-(4-(tert-butyl)phenyl)-5,5-dimethyl-2-oxo-2,5-dihydro**-furan-3-yl)**ethylidene)amino)oxy)methyl)phenyl)-2-(methoxyimino)acetate* (**V**-**9**): White solid, Rf 0.52 (*V*_ethyl acetate_: *V*_petroleum ether_ = 1: 3), 0.47 g, yield 28%, m.p.192–193 °C; ^1^H NMR (500 MHz, CDCl_3_) *δ*: 7.39–7.32 (m, 5H), 7.20 (d, *J* = 8.3 Hz, 2H), 7.17–7.13 (m, 1H), 4.93 (s, 2H), 3.99 (s, 3H), 3.78 (s, 3H), 1.88 (s, 3H), 1.58 (s, 3H), 1.33 (s, 9H); ^13^C NMR (126 MHz, CDCl_3_) *δ*: 170.0, 169.2, 163.4, 152.9, 149.6, 149.3, 136.1, 130.0, 129.4, 128.9, 128.5, 128.3, 127.9, 127.8, 125.5, 86.6, 74.6, 63.9, 53.0, 34.9, 31.3, 25.5, 14.8; HR-MS (ESI) *m*/*z*: C_29_H_35_N_2_O_6_ [M + H]^+^, Calcd. 507.2490, Found 507.2491.

*Methyl (E)-2-(2-(((((E)-1-(4-(4-bromophenyl)-5,5-dimethyl-2-oxo-2,5-dihydrofuran-3-yl)ethyli**- dene)amino)oxy)methyl)phenyl)-2-(methoxyimino)acetate* (**V**-**1****0**): White solid, Rf 0.46 (*V*_ethyl acetate_: *V*_petroleum ether_ = 1: 3), 0.16 g, yield 31%, m.p.197–199 °C; ^1^H NMR (500 MHz, CDCl_3_) *δ*: 7.44–7.40 (m, 2H), 7.38 (dd, J = 7.6, 1.0 Hz, 1H), 7.31 (dt, J = 7.6, 1.0 Hz, 1H), 7.15 (d, *J* = 8.0 Hz, 2H), 7.08–7.04 (m, 2H), 4.86 (s, 2H), 4.01 (s, 3H), 3.83 (s, 3H), 1.97 (s, 3H), 1.52 (s, 6H); ^13^C NMR (126 MHz, CDCl_3_) *δ*: 169.7, 167.8, 163.4, 149.6, 148.9, 136.1, 131.7, 129.8, 129.7, 129.4, 128.5, 128.4, 127.8, 123.9, 86.5, 74.6, 63.9, 53.1, 25.3, 14.3; HR-MS (ESI) *m*/*z*: C_25_H_26_BrN_2_O_6_ [M + H]^+^, Calcd. 529.0969, Found 529.0966.

*Methyl (E)-2-(methoxyimino)-2-(2-(((((E)-1-(2-oxo-4-(pyridin-2-yl)-1-oxaspiro [4.5]**dec-3-en-3-yl) ethylidene) amino)oxy)methyl)phenyl)acetate* (**V**-**1****1**): White solid, Rf 0.35 (*V*_ethyl acetate_: *V*_petroleum ether_ = 1: 3), 0.12 g, yield 18%, m.p.146–148 °C; ^1^H NMR (500 MHz, CDCl_3_) *δ*: 8.62 (d, *J* = 4.3 Hz, 1H), 7.45 (td, *J* = 7.8, 1.5 Hz, 1H), 7.39–7.27 (m, 4H), 7.21–7.13 (m, 2H), 4.92 (s, 2H), 3.99 (s, 3H), 3.81 (s, 3H), 2.30–2.18 (m, 2H), 2.03 (s, 3H), 1.86–1.63 (m, 7H), 1.26–1.16 (m, 1H); ^13^C NMR (126 MHz, CDCl_3_) *δ*: 170.5, 166.1, 163.5, 149.7, 149.2, 136.4, 136.3, 129.4, 128.9, 128.5, 127.8, 126.0, 123.9, 89.4, 74.7, 63.9, 53.0, 34.1, 24.6, 22.3, 14.7; HR-MS (ESI) *m*/*z*: C_27_H_30_N_3_O_6_ [M + H]^+^, Calcd. 492.2129, Found 492.2127.

*Methyl (E)-2-(2-(((((E)-1-(5,5-dimethyl-2-oxo-4-(pyridin-2-yl)-2,5-dihydrofuran-3-yl)ethylidene)**amino)oxy)methyl)phenyl)-2-(methoxyimino)acetate* (**V**-**1****2**): White solid, Rf 0.30 (*V*_ethyl acetate_: *V*_petroleum ether_ = 1: 3), 0.13 g, yield 22%, m.p.100–101 °C; ^1^H NMR (500 MHz, CDCl_3_) *δ*: 8.63–8.58 (m, 1H), 7.48–7.27 (m, 5H), 7.20–7.15 (m, 2H), 4.95 (s, 2H), 3.97 (s, 3H), 3.80 (s, 3H), 2.04 (s, 3H), 1.70 (s, 6H); ^13^C NMR (126 MHz, CDCl_3_) *δ*: 170.2, 165.4, 163.4, 149.9, 149.2, 136.3, 129.3, 128.9, 128.5, 127.8, 125.9, 124.1, 87.6, 74.7, 63.8, 52.9, 26.0, 14.7; HR-MS (ESI) *m*/*z*: C_24_H_26_N_3_O_6_ [M + H]^+^, Calcd. 452.1816, Found 452.1814.

*Methyl (E)-2-(methoxyimino)-2-(2-(((((E)-1-(2-oxo-4-(pyridin-4-yl)-1-oxaspiro [4.5]**dec-3-en-3-yl)ethylidene)amino)oxy)methyl)phenyl)acetate* (**V**-**1****3**): Yellow liquid, Rf 0.37 (*V*_ethyl acetate_: *V*_petroleum ether_ = 1: 3), 0.12 g, yield 26%; ^1^H NMR (500 MHz, CDCl_3_) *δ*: 8.54 (d, *J* = 5.9 Hz, 2H), 7.39–7.27 (m, 2H), 7.12 (dd, *J* = 7.5, 1.1 Hz, 1H), 7.07 (d, *J* = 7.0 Hz, 1H), 7.04–7.01 (m, 2H), 4.74 (s, 2H), 3.99 (s, 3H), 3.81 (s, 3H), 2.00 (s, 3H), 1.83–1.53 (m, 9H), 1.10–1.02 (m, 1H); ^13^C NMR (126 MHz, CDCl_3_) *δ*: 169.6, 166.3, 163.3, 149.7, 148.5, 140.2, 135.9, 129.8, 129.5, 128.5, 127.8, 125.4, 122.7, 87.9, 74.6, 63.9, 53.0, 33.5, 24.3, 21.9, 13.8; HR-MS (ESI) *m*/*z*: C_27_H_30_N_3_O_6_ [M + H]^+^, Calcd. 492.2129, Found 492.2128.

*Methyl**(E)-2-(2-(((((E)-1-(5,5-dimethyl-2-oxo-4-(pyridin-4-yl)-2,5-dihydrofuran-3-yl)ethylidene)**amino)oxy)methyl)phenyl)-2-(methoxyimino)acetate* (**V**-**1****4**): Yellow liquid, Rf 0.32 (*V*_ethyl acetate_: *V*_petroleum ether_ = 1: 3), 0.15 g, yield 20%; ^1^H NMR (500 MHz, CDCl_3_) *δ*: 8.55 (d, *J* = 5.0 Hz, 2H), 7.36 (t, *J* = 7.5 Hz, 1H), 7.29 (t, *J* = 7.5 Hz, 1H), 7.14–7.06 (m, 4H), 4.77 (s, 2H), 3.99 (s, 3H), 3.81 (s, 3H), 2.00 (s, 3H), 1.52 (s, 6H); HR-MS (ESI) *m*/*z*: C_24_H_26_N_3_O_6_ [M + H]^+^, Calcd. 452.1816, Found 452.1817.

To a 100 mL three-necked round-bottomed flask, **V**-**1** (0.20 g, 3.94 mmol) and CH_2_Cl_2_ (20 mL) were added, dropped into 30% methylamine aqueous solution (1.3 mL) and stirred for 1 h at 40 °C. The solvent was removed under the reduced pressure, added 20 mL of water, extracted with ethyl acetate (20 × 3 mL) and combined the organic phase. The organic phase was washed with saturated brine, dried over anhydrous Na_2_SO_4_. The solvent was removed under the reduced pressure, and the residue was subjected to silica gel column chromatography and washed with ethyl acetate: petroleum ether to afford a white solid compound **V****I**-**1**. Compounds **V**-**2**–**V**-**14** were synthesized in a similar protocol.

*(E)-2-(2-(((((E)-1-(4-(4-**Fluorophenyl)-2-oxo-1-oxaspiro[4.5]dec-3-en-3-yl)ethylidene)amino)oxy)**methyl)phenyl)-2-(methoxyimino)-N-methylacetamide* (**V****I**-**1**): White solid, Rf 0.41 (*V*_ethyl acetate_: *V*_petroleum ether_ = 1: 2), 0.18 g, yield 75%, m.p.70–73 °C; ^1^H NMR (500 MHz, CDCl_3_) *δ*: 7.34 (td, *J* = 7.5, 1.1 Hz, 1H), 7.27 (dt, *J* = 7.5, 1.1 Hz, 1H), 7.20–7.07 (m, 4H), 7.01–6.95 (m, 2H), 6.76 (q, *J* = 5.0 Hz, 1H), 4.86 (s, 2H), 3.92 (s, 3H), 2.88 (d, *J* = 5.0 Hz, 3H), 1.90 (s, 3H), 1.80–1.57 (m, 9H), 1.13–1.05 (m, 1H); ^13^C NMR (126 MHz, CDCl_3_) *δ*: 170.2, 168.9, 163.1 (d, ^1^*J*_FC_ = 247.6 Hz), 163.0, 151.3, 148.8, 136.2, 130.1 (d, ^3^*J*_FC_ = 8.4 Hz), 129.5, 129.2, 128.7, 128.4, 127.6, 127.5 (d, ^4^*J*_FC_ = 3.4 Hz), 125.4, 115.6 (d, ^2^*J*_FC_ = 21.4 Hz), 88.3, 74.7, 63.3, 33.6, 26.4, 24.5, 22.1, 14.4; HR-MS (ESI) *m*/*z*: C_28_H_31_FN_3_O_5_ [M + H]^+^, Calcd. 508.2242, Found 5098.2245.

*(E)-2-(**Methoxyimino)-2-(2-(((((E)-1-(4-(4-methoxyphenyl)-2-oxo-1-oxaspiro[4.5]dec-3-en-3-yl)ethylidene)amino)oxy)methyl)phenyl)-N-methylacetamide* (**V****I**-**2**): White solid, Rf 0.46 (*V*_ethyl acetate_: *V*_petroleum ether_ = 1: 2), 0.16 g, yield 80%, m.p.56–60 °C; ^1^H NMR (500 MHz, CDCl_3_) *δ*: 7.37–7.28 (m, 2H), 7.25 (dd, *J* = 8.0, 1.5 Hz, 1H), 7.18 (dd, *J* = 7.5, 1.5 Hz, 1H), 7.15–7.11 (m, 2H), 6.88–6.83 (m, 2H), 6.76 (q, *J* = 5.0 Hz, 1H), 4.95 (s, 2H), 3.92 (s, 3H), 3.83 (s, 3H), 2.86 (d, *J* = 5.0 Hz, 3H), 1.82 (s, 3H), 1.81–1.65 (m, 9H), 1.19–1.05 (m, 1H); ^13^C NMR (126 MHz, CDCl_3_) *δ*: 170.4, 169.8, 163.1, 160.5, 151.3, 149.1, 136.1, 129.8, 129.6, 129.2, 128.7, 128.7, 127.7, 124.6, 123.6, 114.0, 88.4, 74.8, 63.3, 55.4, 33.8, 26.4, 24.6, 22.1, 14.8; HR-MS (ESI) *m*/*z*: C_29_H_34_N_3_O_6_ [M + H]^+^, Calcd. 520.2242, Found 520.2240.

*(E)-2-(2-(((((E)-1-(4-(4-(tert-**Butyl)phenyl)-2-oxo-1-oxaspiro[4.5]dec-3-en-3-yl)ethylidene)amino)**oxy)methyl)phenyl)-2-(methoxyimino)-N-methylacetamide* (**V****I**-**3**): White solid, Rf 0.45 (*V*_ethyl acetate_: *V*_petroleum ether_ = 1: 2), 0.21 g, yield 87%, m.p.62–65 °C; ^1^H NMR (500 MHz, CDCl_3_) *δ*: 7.40–7.28 (m, 5H), 7.22–7.17 (m, 1H), 7.16–7.11 (m, 2H), 6.68 (q, *J* = 5.0 Hz, 1H), 4.94 (s, 2H), 3.91 (s, 3H), 2.82 (d, *J* = 5.0 Hz, 3H), 1.84–1.67 (m, 13H), 1.33 (s, 9H), 1.14–1.10 (m, 1H); ^13^C NMR (126 MHz, CDCl_3_) *δ*: 170.3, 170.1, 163.1, 152.6, 151.4, 149.0, 135.9, 129.3, 128.9, 128.8, 128.7, 127.8, 125.4, 88.4, 75.0, 63.3, 34.9, 33.7, 31.4, 26.4, 24.5, 22.1, 14.9; HR-MS (ESI) *m*/*z*: C_32_H_40_N_3_O_5_ [M + H]^+^, Calcd. 546.2966, Found 546.2962.

*(E)-2-(2-(((((E)-1-(4-(4-**Bromophenyl)-2-oxo-1-oxaspiro[4.5]dec-3-en-3-yl)ethylidene)amino)oxy)**methyl)phenyl)-2-(methoxyimino)-N-methylacetamide* (**V****I**-**4**): White solid, Rf 0.42 (*V*_ethyl acetate_: *V*_petroleum ether_ = 1: 2), 0.24 g, yield 82%, m.p.130–131 °C; ^1^H NMR (500 MHz, CDCl_3_) *δ*: 7.41 (d, *J* = 8.5 Hz, 2H), 7.37 (t, *J* = 7.5 Hz, 1H), 7.29 (t, *J* = 7.5 Hz, 1H), 7.16 (d, *J* = 7.5 Hz, 1H), 7.12 (d, *J* = 7.5 Hz, 1H), 7.00 (d, *J* = 8.5 Hz, 2H), 6.75 (q, *J* = 5.0 Hz, 1H), 4.85 (s, 2H), 3.92 (s, 3H), 2.89 (d, *J* = 5.0 Hz, 3H), 1.92 (s, 3H), 1.81–1.62 (m, 9H), 1.13–1.02 (m, 1H); ^13^C NMR (126 MHz, CDCl_3_) *δ*: 170.1, 168.5, 163.0, 151.2, 148.7, 136.1, 131.6, 130.5, 129.7, 129.5, 129.3, 128.7, 128.4, 127.6, 125.3, 123.6, 88.2, 74.7, 63.4, 33.6, 26.4, 24.4, 22.0, 14.4; HR-MS (ESI) *m*/*z*: C_28_H_31_BrN_3_O_5_ [M + H]^+^, Calcd. 568.1442, Found 568.1446.

*(E)-2-(2-(((((E)-1-(4-(2-**Chlorophenyl)-2-oxo-1-oxaspiro[4.5]dec-3-en-3-yl)ethylidene)amino)oxy)**methyl)phenyl)-2-(methoxyimino)-N-methylacetamide* (**V****I**-**5**): White solid, Rf 0.51 (*V*_ethyl acetate_: *V*_petroleum ether_ = 1: 2), 0.19 g, yield 85%, m.p.65–67 °C; ^1^H NMR (500 MHz, CDCl_3_) *δ*: 7.39 (dd, *J* = 7.7, 1.5 Hz, 1H), 7.33–7.24 (m, 4H), 7.17–7.05 (m, 3H), 6.67 (q, *J* = 5.0 Hz, 1H), 4.71 (s, 2H), 3.90 (s, 3H), 2.88 (d, *J* = 5.0 Hz, 3H), 1.99 (s, 3H), 1.94–1.65 (m, 8H), 1.54–1.47 (m, 1H), 1.11–1.05 (m, 1H); ^13^C NMR (126 MHz, CDCl_3_) *δ*: 169.9, 166.1, 162.9, 151.2, 148.7, 135.9, 131.3, 130.0, 129.9, 129.7, 129.5, 129.3, 128.7, 128.5, 127.6, 126.4, 88.9, 74.8, 63.3, 33.7, 33.5, 26.4, 24.4, 22.1, 22.0 13.1; HR-MS (ESI) *m*/*z*: C_28_H_31_ClN_3_O_5_ [M + H]^+^, Calcd. 524.1947, Found 524.1949.

*(E)-2-(2-(((((E)-1-(5,5-**Dimethyl-2-oxo-4-phenyl-2,5-dihydrofuran-3-yl)ethylidene)amino)oxy)**methyl)phenyl)-2-(methoxyimino)-N-methylacetamide* (**V****I**-**6**): Yellow liquid, Rf 0.42 (*V*_ethyl acetate_: *V*_petroleum ether_ = 1: 2), 0.25 g, yield 81%; ^1^H NMR (500 MHz, CDCl_3_) *δ*: 7.42–7.28 (m, 5H), 7.25–7.20 (m, 3H), 7.17 (dd, *J* = 7.5, 1.0 Hz, 1H), 6.73 (q, *J* = 5.0 Hz, 1H), 4.91 (s, 2H), 3.91 (s, 3H), 2.84 (d, *J* = 5.0 Hz, 3H), 1.84 (s, 3H), 1.55 (s, 6H); ^13^C NMR (126 MHz, CDCl_3_) *δ*: 169.8, 169.3, 163.0, 151.3, 148.8, 135.9, 131.3, 129.8, 129.6, 129.3, 128.7, 128.7, 128.6, 127.9, 127.7, 86.6, 74.9, 63.3, 26.3, 25.3, 14.6; HR-MS (ESI) *m*/*z*: C_25_H_28_N_3_O_5_ [M + H]^+^, Calcd. 450.2023, Found 450.2026.

*(E)-2-(2-(((((E)-1-(4-(4-**Fluorophenyl)-5,5-dimethyl-2-oxo-2,5-dihydrofuran-3-yl)ethylidene)**amino)oxy)methyl)phenyl)-2-(methoxyimino)-N-methylacetamide* (**V****I**-**7**): Yellow liquid, Rf 0.40 (*V*_ethyl acetate_: *V*_petroleum ether_ = 1: 2), 0.21 g, yield 84%; ^1^H NMR (500 MHz, CDCl_3_) *δ*: 7.35 (t, *J* = 7.5 Hz, 1H), 7.28 (t, *J* = 7.5 Hz, 1H), 7.22–7.13 (m, 4H), 6.98 (t, *J* = 8.5 Hz, 2H), 6.79 (q, *J* = 5.0 Hz, 1H), 4.90 (s, 2H), 3.92 (s, 3H), 2.89 (d, *J* = 5.0 Hz, 3H), 1.92 (s, 3H), 1.54 (s, 6H); ^13^C NMR (126 MHz, CDCl_3_) *δ*: 169.8, 168.2, 163.3 (d, ^1^*J*_FC_ = 248.4 Hz), 163.0, 151.3, 148.8, 136.2, 130.1 (d, ^3^*J*_FC_ = 8.1 Hz), 129.6, 129.2, 128.8, 128.5, 127.6, 127.1 (d, ^4^*J*_FC_ = 3.4 Hz), 124.8, 115.7 (d, ^2^*J*_FC_ = 21.8 Hz), 86.5, 74.8, 63.3, 26.4, 25.3, 14.4; HR-MS (ESI) *m*/*z*: C_25_H_27_FN_3_O_5_ [M + H]^+^, Calcd. 468.1929, Found 468.1927.

*(E)-2-(**Methoxyimino)-2-(2-(((((E)-1-(4-(4-methoxyphenyl)-5,5-dimethyl-2-oxo-2,5-dihydrofura**n -3-yl)ethylidene)amino)oxy)methyl)phenyl)-N-methylacetamide* (**V****I**-**8**): White solid, Rf 0.36 (*V*_ethyl acetate_: *V*_petroleum ether_ = 1: 2), 0.15 g, yield 88%, m.p.160–161 °C; ^1^H NMR (500 MHz, CDCl_3_) *δ*: 7.39–7.27 (m, 3H), 7.22 (d, *J* = 8.8 Hz, 2H), 7.20 (dd, *J* = 7.0, 1.5 Hz, 1H), 6.86 (d, *J* = 8.8 Hz, 2H), 6.79 (q, *J* = 5.0 Hz, 1H), 4.98 (s, 2H), 3.93 (s, 3H), 3.84 (s, 3H), 2.86 (d, *J* = 5.0 Hz, 3H), 1.87 (s, 3H), 1.58 (s, 6H); ^13^C NMR (126 MHz, CDCl_3_) *δ*: 170.1, 168.8, 163.2, 160.8, 151.4, 149.3, 136.1, 129.9, 129.8, 129.3, 128.8, 127.7, 123.2, 114.2, 86.6, 74.9, 63.4, 55.5, 26.4, 25.7, 14.9; HR-MS (ESI) *m*/*z*: C_26_H_30_N_3_O_6_ [M + H]^+^, Calcd. 480.2131, Found 480.21929.

*(E)-2-(2-(((((E)-1-(4-(4-(tert-**Butyl)phenyl)-5,5-dimethyl-2-oxo-2,5-dihydrofuran-3-yl)ethylidene)**amino)oxy)methyl)phenyl)-2-(methoxyimino)-N-methylacetamide* (**V****I**-**9**): White solid, Rf 0.49 (*V*_ethyl acetate_: *V*_petroleum ether_ = 1: 2), 0.22 g, yield 81%, m.p.194–195 °C; ^1^H NMR (500 MHz, CDCl_3_) *δ*: 7.40–7.31 (m, 5H), 7.21–7.15 (m, 3H), 6.71 (q, *J* = 5.0 Hz, 1H), 4.97 (s, 2H), 3.92 (s, 3H), 2.82 (d, *J* = 5.0 Hz, 3H), 1.82 (s, 3H), 1.60 (s, 3H), 1.33 (s, 9H); ^13^C NMR (126 MHz, CDCl_3_) *δ*: 170.0, 169.2, 163.1, 153.2, 151.4, 149.1, 136.0, 130.1, 129.3, 129.0, 128.9, 128.2, 128.0, 127.9, 125.6, 86.6, 75.1, 63.4, 34.9, 31.3, 26.4, 25.6, 14.9; HR-MS (ESI) *m*/*z*: C_29_H_36_N_3_O_5_ [M + H]^+^, Calcd. 506.6249, Found 506.6247.

*(E)-2-(2-(((((E)-1-(4-(4-**Bromophenyl)-5,5-dimethyl-2-oxo-2,5-dihydrofuran-3-yl)ethylidene)**amino)oxy)methyl)phenyl)-2-(methoxyimino)-N-methylacetamide* (**V****I**-**1****0**): White solid, Rf 0.49 (*V*_ethyl acetate_: *V*_petroleum ether_ = 1: 2), 0.16 g, yield 80%, m.p.188–190 °C; ^1^H NMR (500 MHz, CDCl_3_) *δ*: 7.44–7.35 (m, 3H), 7.29 (td, *J* = 7.5, 1.0 Hz, 1H), 7.17 (dd, *J* = 7.8, 1.0 Hz, 1H), 7.14 (d, *J* = 7.5 Hz, 1H), 7.08–7.05 (m, 2H), 6.77 (q, *J* = 5.0 Hz, 1H), 4.88 (s, 2H), 3.93 (s, 3H), 2.89 (d, *J* = 5.0 Hz, 3H), 1.94 (s, 3H), 1.53 (s, 6H); ^13^C NMR (126 MHz, CDCl_3_) *δ*: 169.8, 167.8, 163.0, 151.3, 148.8, 131.8, 130.1, 129.7, 129.3, 128.8, 128.5, 127.7, 124.0, 86.5, 74.8, 63.4, 26.4, 25.3, 14.4; HR-MS (ESI) *m*/*z*: C_25_H_27_BrN_3_O_5_ [M + H]^+^, Calcd. 528.1129, Found 528.1126.

*(E)-2-(**Methoxyimino)-**N-methyl-2-(2-(((((E)-1-(2-oxo-4-(pyridin-2-yl)-1-oxaspiro [4.5]dec-3-en-3-yl)ethylidene)amino)oxy)methyl)phenyl)acetamide* (**V****I**-**1****1**): White solid, Rf 0.32 (*V*_ethyl acetate_: *V*_petroleum ether_ = 1: 1), 0.14 g, yield 88%, m.p.139–140 °C; ^1^H NMR (500 MHz, CDCl_3_) *δ*: 8.62–8.59 (m, 1H), 7.44 (td, *J* = 7.8, 1.8 Hz, 1H), 7.37–7.26 (m, 3H), 7.25–7.15 (m, 3H), 6.82 (q, *J* = 5.0 Hz, 1H), 4.95 (s, 2H), 3.92 (s, 3H), 2.89 (d, *J* = 5.0 Hz, 3H), 2.26 (brs, 1H), 1.99 (s, 3H), 1.84–1.62 (m, 8H), 1.26–1.17 (m, 1H); ^13^C NMR (126 MHz, CDCl_3_) δ: 170.6, 166.2, 163.1, 149.5, 149.2, 136.5, 130.0, 129.2, 128.9, 128.8, 127.7, 126.1, 125.4, 124.0, 89.4, 75.0, 63.4, 34.1, 26.4, 24.6, 22.3, 14.8; HR-MS (ESI) *m*/*z*: C_27_H_31_N_4_O_5_ [M + H]^+^, Calcd. 491.2291, Found 491.2286.

*(E)-2-(2-(((((E)-1-(5,5-**Dimethyl-2-oxo-4-(pyridin-2-yl)-2,5-dihydrofuran-3-yl)ethylidene)amino)**oxy)methyl)phenyl)-2-(methoxyimino)-N-methylacetamide* (**V****I**-**1****2**): White solid, Rf 0.31 (*V*_ethyl acetate_: *V*_petroleum ether_ = 1: 1), 0.16 g, yield 86%, m.p.146–147 °C; ^1^H NMR (500 MHz, CDCl_3_) *δ*: 8.63–8.58 (m, 1H), 7.49–7.33 (m, 3H), 7.32–7.25 (m, 2H), 7.24–7.15 (m, 2H), 6.88 (q, *J* = 5.0 Hz, 1H), 4.99 (s, 2H), 3.93 (s, 3H), 2.89 (d, *J* = 5.0 Hz, 3H), 2.04 (s, 3H), 1.72 (s, 6H); ^13^C NMR (126 MHz, CDCl_3_) *δ*: 170.3, 165.5, 163.1, 151.3, 149.9, 149.6, 149.1, 136.5, 130.1, 129.2, 129.0, 128.9, 127.7, 126.0, 124.9, 124.2, 87.7, 75.0, 63.3, 26.3, 26.1, 14.8; HR-MS (ESI) *m*/*z*: C_24_H_27_N_4_O_5_ [M + H]^+^, Calcd. 451.1976, Found 451.1978.

*(E)-2-(**Methoxyimino)-N-methyl-2-(2-(((((E)-1-(2-oxo-4-(pyridin-4-yl)-1-oxaspiro [4.5]dec-3-en-3-yl) ethylidene)amino)oxy)methyl)phenyl)acetamide* (**V****I**-**1****3**): White solid, Rf 0.33 (*V*_ethyl acetate_: *V*_petroleum ether_ = 1: 1), 0.18 g, yield 87%, m.p.119–120 °C; ^1^H NMR (500 MHz, CDCl_3_) *δ*: 8.54–8.52 (m, 2H), 7.34 (td, *J* = 7.5, 1.0 Hz, 1H), 7.28 (td, *J* = 7.5, 1.0 Hz, 1H), 7.15 (dd, *J* = 7.5, 1.5 Hz, 1H), 7.06 (d, *J* = 7.0 Hz, 1H), 7.03–7.01 (m, 2H), 6.75 (q, *J* = 5.0 Hz, 1H), 4.78 (s, 2H), 3.91 (s, 3H), 2.89 (d, *J* = 5.0 Hz, 3H), 1.97 (s, 3H), 1.83–1.54 (m, 9H), 1.13–1.02 (m, 1H); ^13^C NMR (126 MHz, CDCl_3_) *δ*: 169.7, 166.3, 162.9, 151.2, 149.7, 148.4, 140.2, 136.1, 129.5, 129.4, 128.8, 128.3, 127.7, 125.6, 122.7, 87.9, 74.8, 63.4, 33.5, 26.4, 24.4, 22.0, 14.0; HR-MS (ESI) *m*/*z*: C_27_H_31_N_4_O_5_ [M + H]^+^, Calcd. 491.2291, Found 491.2289.

*(E)-2-(2-(((((E)-1-(5,5-**Dimethyl-2-oxo-4-(pyridin-4-yl)-2,5-dihydrofuran-3-yl)ethylidene)amino)**oxy) methyl)phenyl)-2-(methoxyimino)-N-methylacetamide* (**V****I**-**1****4**): White solid, Rf 0.32 (*V*_ethyl acetate_: *V*_petroleum ether_ = 1: 1), 0.15 g, yield 85%, m.p.162–163 °C; ^1^H NMR (500 MHz, CDCl_3_) *δ*: 8.53 (d, *J* = 6.0 Hz, 2H), 7.33 (t, *J* = 7.5 Hz, 1H), 7.27–7.24 (m, 1H), 7.14 (d, *J* = 7.5 Hz, 1H), 7.09–7.04 (m, 3H), 6.78 (q, *J* = 5.0 Hz, 1H), 4.79 (s, 2H), 3.89 (s, 3H), 2.88 (d, *J* = 5.0 Hz, 3H), 1.97 (s, 3H), 1.51 (s, 6H); ^13^C NMR (126 MHz, CDCl_3_) *δ*: 169.3, 165.7, 162.9, 151.1, 149.9, 148.3, 139.5, 136.0, 129.5, 129.3, 128.7, 128.3, 127.6, 125.4, 122.5, 86.2, 74.8, 63.3, 26.3, 25.0, 13.9; HR-MS (ESI) *m*/*z*: C_24_H_27_N_4_O_5_ [M + H]^+^, Calcd. 451.1976, Found 451.1971.

### 3.4. Determination of the In Vitro Antifungal Activity of Compounds

The in vitro antifungal activity evaluation: the phytopathogens *F. graminearum*, *S. sclerotiorum*, *B. cinerea*, *P. capsici* and *T. cucumeris* used in this paper were isolated and preserved by the Department of Plant Pathology, College of Plant Protection, China Agricultural University, and further activation was required for the activity test. The mycelium growth rate method was used for the determination in potato dextrose agar (PDA), as described: the stock 2000 µg/mL DMSO solution of tested compounds were prepared in advance. Then, hot culture medium (9.75 mL) was added into a plate, added sample solution (0.25 mL) or blank DMSO (0.25 mL) to the plate and mix with PDA culture medium, making the final concentration 50 µg/mL. When plate was made, we put a 5 mm diameter fungus cake onto the center of plate, incubated them at 25 ± 0.5 °C for 24–168 h, and the commercial fungicide TRI was used as the control agent, and the experiments were triplicates for each treatment. The experiments were run with 9 cm Petri dish, the mycelium diameters were determined after 2–7 days for blank control and treatment with 50 mg/L compounds, which depend on the different phytopathogens. After the initial screening, 7 concentration gradients (50, 25, 12.5, 6.25, 3.13, 1.56, 0.78 mg/L) were set for the compounds with good antifungal activity to determine the EC_50_ values, and SPSS Statistics 25 software was used to determine the EC_50_ value.

*S. sclerotiorum* was cultured in PDA using 12.5 mg/L **VI-3** and TRI for 3 days at 25 °C, the mycelial tips (5 mm) of an actively growing colony on PDA medium amended 12.5 mg/L compounds **VI-3** and TRI were cut from the edge of the colony cultured for 72 h. The tips were treated with 2% glutaraldehyde at 4 °C, followed by rinsing with 0.1 mol/L phosphate buffer (pH 7.3) and fixed with 1.0 g/mL osmium tetraoxide solution. After rinsed with 0.1 mol/L phosphate buffer three times, the mycelial tips were dehydrated using a series of ethanol solutions in the order of concentration 30%, 50%, 70%, 80%, 90% and 100%. The processes of drying at critical point, mounting and gold spraying were completed at last and examined with a Cold Field-Emission SU-8010 scanning electron microscope (SEM) as with the previous paper [41].

The mycelial tips were prepared according to the method given above. After dehydrating with acetone and embedding in SPURR resin, thin sections were cut with LEICAUCi machine and double-stained with uranyl acetate and lead citrate. The grids were examined with a Hitachi-7500 transmission electron microscope (TEM), as with the previous paper [41].

## 4. Conclusions

In summary, a series of novel butenolides containing methoxyacrylate moiety were designed and synthesized, and their antifungal activities against phytopathogens were evaluated. The preliminary results showed that the inhibitory activities of these new compounds against *S. sclerotiorum* were significantly improved compared with that of lead **3–8**. The EC_50_ values of **V**-**6** and **V****I**-**7** against *S. sclerotiorum* were 1.51 and 1.81 mg/L, respectively, nearly seven times higher than that of **3–8** (EC_50_ = 10.62 mg/L). The transmission electron microscopy and scanning electron microscopy observation of the hyphae and cell morphology of *S. sclerotiorum* showed that TRI and **V**-**3** should have similar mechanism of action, act as the inhibitor of mitochondrial respiration system by blocking electron transfer at the ubiquinol oxidation center (Qo site) of the cytochrome bc1 complex. Molecular docking results indicated that the introduction of methoxyacrylate scaffold is beneficial to improve the fungicidal activities of these compounds due to the hydrogen bond of the key Glu 271 amino acid residue in the active cavity of the protein with title compounds **V**-**6** and **V****I**-**7**. Compounds **V**-**6** and **V****I**-**7** can be used as the new leads to optimize their structures; the further modification is running in our laboratory.

## Data Availability

Not applicable.

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
