# Peer review of "Synthesis and Antifungal Activity of New butenolide Containing Methoxyacrylate Scaffold"

_molecules, 2022, doi:10.3390/molecules27196541_

Round 1
Reviewer 1 Report
In this manuscript, the authors reported a new butenolide compounds bearing methoxyacrylate scaffold and evaluated their in vitro fungicidal activities against phytopathogens. Also, authors claim that methoxyacrylate improve the fungicidal activity based on molecular docking and TEM and SEM analysis shown one of the scaffolds had a significant impact on the structure and function of the hyphal cell wall of one of the species as the positive control such as TRI. Overall, I regret to say I was not impressed by the work, moreover by the authors interpretation of the results and lock of novelty. The numbering of formulas in the Schemes are chaotic and it is very uncomfortable to readers (starting they mentioned 1, 2, 3 and different one A8).
Comments
1. In abstract, novel should be changed as new and uniform as well as no specific numbering given in the abstract.
2. In introduction, QoI means?? Second paragraph should rephrase it (line 63-68) and difficult to digest content of the concept.
3. The schemes 1 to 2 should be one scheme and comparison of results (previous/present) explained in detailed manner including proper citations.
4. In scheme 3, authors should provide experimental conditions in detail such as time time/temp.etc…and numbering should be modified.
5. In exp.section, weights of the products and Rf values must be included.
6. Typos should be minimized, for ex..line 176 phytopathogen…etc.. and repeated words for example In addition in line 141 and 144 should be changed etc… as well as references styles double checked again ..reference 42 synthesis should be italics…
7. In SI info: 1C should be 13C…NMR data not consistent with the provided NMR copies and rechecked once for example compound b7 13C peak at 164.2 but in data no info.. and solvent information/NMR nuclei should be mentioned for each case/NMR window should be 10 ppm for proton and 210 ppm for carbon should be uniform in all cases. I can see that several places some unusual peaks and traces of impurities should be considered.
Reviewer 2 Report
The manuscript in reference describes the synthesis and antifungal activity against five fungal species of a set of butenolide-containing compounds. The manuscript is well-written and has relevant results and interesting information for the readership. However, some minor points should be addressed prior to further consideration.
1. Detailed scrutiny is recommended throughout the manuscript to revise/correct some grammar, stylistic issues, and even typos.
2. Title and other parts of the manuscript: I am not convinced about the fungicidal term since this study did not confirm the fungicidal effect, and possibly, compounds could have a fungistatic effect. According to the information presented in the manuscript, I suggest using the term “antifungal” instead of “fungicidal”. The antifungal term is suitable for the effect of the test compounds evaluated by the authors through the mycelium growth inhibition assay. Be consistent throughout the manuscript.
3. Lines 9-12: revise this passage since it is too long and challenging to be followed.
4. Lines 12-133: the employed antifungal method can be mentioned here.
5. Line 47: “Natural products” instead of “nature products”
6. Lines 48-53: Can butenolide moiety be considered a toxophore? Or pro-antifungal? This information can be expanded and clarified in this passage as a hypothesis to support the literature survey.
7. Lines 63-68: revise this passage since it is too long and challenging to be followed.
8. Line 70: Add the reference for the caption of this Scheme 1.
9. Lines 116 and 151: Unify the dose presentation (i.e., mg/L in lines116/151 or µg/mL in abstract).
10. Line 139: Specify how much were smaller? An inferential test is also suggested.
11. Line 142: Specify how much was improved? An inferential test is also suggested.
12. Lines 144-148: The effects evaluated by TEM must be expanded and discussed adequately for readers.
13. Lines 149-150: This affirmation is not clear. I suggest expanding and comparing in detail with other reported studies to support such a deductive statement.
14. Table 1: Inferential statistics (e.g., multiple comparisons) to define significant differences must be performed in this dataset per fungus. In addition, specify what type of interval is related after ±? Standard deviation? SEM? Indicate also the number of replicates.
15. Figure 4: The images of this figure should be adequately enlarged since the text information is difficult to read.
16. Conclusions comprise a summary of the results. I recommend re-writing conclusions into specific conceptual findings from the mechanistic point of view and, subsequently, including the scope of this information for future studies.
17. Revise in detail the M&M section. Some experimental details are missing to ensure outcome reproducibility. For instance, brand, model, and grade of reagents, solvents, materials, and instruments must be provided.
18. Line 208: 13C NMR data should be expressed with just one decimal figure. Be consistent throughout the manuscript.
19. Line 518: More details about antifungal activity evaluation must be provided, such as dose range, culture medium, inoculum, the incubation (time and temperature), response (diameter or area of growth), scale (Petri dishes or smaller wells), etc. As written, it is not possible to ensure reproducibility.
20. Line 528: Details for SEM preparation are also missing and must be adequately informed.
Round 2
Reviewer 1 Report
suggested concerns addressed by authors reasonably and happy to recommend it for molecules.